# Learning to Parallel: Accelerating Diffusion Large Language Models via Learnable Parallel Decoding

**Wenrui Bao**[1*], **Zhiben Chen**[2*], **Dan Xu**[3], **Yuzhang Shang**[1†]
[1]University of Central Florida, [2]Mobi.AI, [3]HKUST
we769358@ucf.edu; zhiben@mobi.ai

## Abstract

Autoregressive decoding in large language models (LLMs) requires $\mathcal{O}(n)$ sequential steps for $n$ tokens, fundamentally limiting inference throughput. Recent diffusion-based LLMs (dLLMs) enable parallel token generation through iterative denoising. However, current parallel decoding strategies rely on fixed, input-agnostic heuristics (e.g., confidence thresholds), which fail to adapt to input-specific characteristics, resulting in suboptimal speed-quality trade-offs across diverse NLP tasks. In this work, we explore a more flexible and dynamic approach to parallel decoding. We propose **Learning to Parallel Decode (Learn2PD)**, a framework that trains a lightweight and adaptive filter model to predict, for each token position, whether the current prediction matches the final output. This learned filter approximates an oracle parallel decoding strategy that unmasks tokens only when correctly predicted. Importantly, the filter model is learned in a post-training manner, requiring only a small amount of computation to optimize it (minute-level GPU time). Additionally, we introduce **End-of-Text Prediction (EoTP)** to detect decoding completion at the end of sequence, avoiding redundant decoding of padding tokens. Experiments on the LLaDA [Nie et al., 2025] benchmark demonstrate that our method achieves up to **22.58×** speedup without any performance drop, and up to **57.51×** when combined with KV-Cache. To facilitate reproducibility and future research, the project page and code are publicly available at Project Page and GitHub Repository.

## 1 Introduction

Large Language Models (LLMs) [Zhao et al., 2023, Ziyu et al., 2023, Minaee et al., 2024] have demonstrated remarkable capabilities across a wide spectrum of natural language processing (NLP) tasks. However, most state-of-the-art LLMs rely on autoregressive (AR) decoding [Brown et al., 2020, Radford et al., 2019, Vaswani et al., 2017], which generates output tokens sequentially. Although this approach delivers strong generation quality, it inherently suffers from limited inference efficiency due to its strictly sequential nature [Leviathan et al., 2023, Stern et al., 2018]. To overcome this bottleneck, diffusion-based LLMs (dLLMs) [Nie et al., 2025, Ye et al., 2025] have been proposed as a compelling alternative by enabling parallel token generation through iterative denoising, potentially achieving sublinear complexity [Sohl-Dickstein et al., 2015, Li et al., 2022].

Diffusion-based LLMs (dLLMs) produce or iteratively refine the entire token sequence via denoising steps rather than predicting tokens one by one, so token-wise predictions at each step can be computed in parallel. Especially, most dLLMs adopt *semi-autoregressive decoding* [Arriola et al., 2025], which divides the target sequence into contiguous blocks and decodes the blocks from left to right. It facilitates token-parallelism by trading a small amount of autoregressive constraint for substantially higher parallel throughput, while still preserving essential left-to-right dependencies. To fully unlock these benefits, further development of a parallel decoding strategy that can leverage this approach is needed. Current methods employ static heuristics, for example, *confidence-based*

---

[*]Equal contribution.
[†]Corresponding author.

Figure 1: **Effectiveness of our proposed approaches.** We report the throughput and accuracy on GSM8K (5-shot, Generation Length=1024) with LLaDA and our proposed methods under four settings: (1) vanilla decoding, (2) Learn2PD policy, (3) Learn2PD and EoTP mechanism, (4) Learn2PD and EoTP integrated by KV Cache. Our proposed methods, Learn2PD and EoTP, yield a $22.58\times$ speedup over the vanilla baseline while simultaneously preserving the original accuracy. Integration with KV Cache achieves a further improvement in throughput to 16.37 tokens/sec (a $57.51\times$ speedup), with only a minimal loss in accuracy.

*sampling* [Chang et al., 2022] prioritizes the most confident tokens for parallel decoding. Although these methods speed up inference, their static decoding strategies lead to poor generation quality.

Targeting this static limitation, we pose an intuitive question: *Instead of relying on a one-rule-fits-all decoding strategy, can we adopt a flexible, case-by-case one for parallel decoding?* To answer this, we analyzed the model's token-level decoding behavior and found that current models often remask tokens that have already been correctly predicted, leading to unnecessary computational redundancy. Taking advantage of this finding, we propose that an effective parallel decoding strategy should be capable of eliminating such redundancy. To realize this goal, we first establish an oracle baseline: **Extremely Greedy Parallel (EGP)**, which unmasks each token immediately upon correct prediction. In the oracle, we use the reference answers to unmask a token when its prediction matches the ground truth. Our analysis reveals that this oracle can achieve a 15-20× speedup without quality loss, demonstrating substantial potential to improve parallel decoding. However, its dependence on unavailable ground truth makes it infeasible in practice.

To approximate this oracle, we propose **Learning to Parallel Decode (Learn2PD)**, the first learned parallel decoding policy for dLLMs. The framework learns to predict when to finalize a token—that is, when we have sufficient confidence to accept its current prediction. The key insight is that diffusion models exhibit predictable confidence patterns [Song et al., 2020, Nichol & Dhariwal, 2021]: the confidence score for each token can be treated as an informative feature. Fluctuations in these scores capture the model's internal state of acceptance or doubt regarding its predictions. Specifically, we train a lightweight filter model $f_\theta$ that predicts whether each token has been correctly generated. The filter model is optimized in the post-training phase, requiring minute-level GPU time for convergence. Once trained, this filter model remains fixed and requires no gradient updates during inference. The filter takes the model's confidence scores as input and outputs a binary decision for each token to indicate whether it should be remasked. Surprisingly, a simple two-layer MLP [Tolstikhin et al., 2021] performs exceptionally well at this task, as the block-level confidence patterns provide sufficient information for accurate convergence prediction, thus eliminating the need for complex architectures or task-specific feature engineering.

Another finding from the EGP oracle is that even when the [End-of-Text] token is unmasked, the model continues the decoding process for subsequent tokens. When the generation length is 1024, this inefficiency is responsible for 90% of the computational waste. To reduce the excessive decoding steps after the [End-of-Text] token, we introduce an **End-of-Text Prediction (EoTP)** mechanism. EoTP can terminate decoding as soon as the [End-of-Text] token is confidently generated, which avoids redundant computation and further boosts decoding efficiency.

Our method accelerates dLLMs by eliminating redundant decoding operations, thereby preserving generation quality. Experimental results demonstrate a remarkable **22.58×** speed-up on LLaDA while fully maintaining its performance. Importantly, our method is **orthogonal** to existing optimizations: when combined with KV caching, the speedup compounds to **57.51×** accompanied by only a slight degradation in accuracy (See Figure 1). In summary, our contributions are threefold:

1. We propose a novel and adaptive framework, **Learn2PD** that predicts which tokens have been correctly decoded, approximating the oracle Extremely Greedy Parallel Decoding strategy.

2. We also propose a **End-of-Text Prediction (EoTP)** mechanism to reduce the unnecessary decoding steps, which significantly boosts inference efficiency.

3. We extensively evaluate our method on various dLLMs across four representative benchmarks: GSM8K, MATH, HumanEval, and MBPP. Our method consistently achieves order-of-magnitude inference acceleration with negligible accuracy loss. Specifically, our method attains a significant **22.58×** acceleration without any degradation in accuracy.

## 2 RELATED WORK

### 2.1 DIFFUSION-BASED LARGE LANGUAGE MODELS

The integration of diffusion models with large language models (LLMs) is an emerging and promising direction in generative AI. Early work adapted continuous diffusion to discrete data domains [Sohl-Dickstein et al., 2015, Hoogeboom et al., 2021], leading to D3PM [Austin et al., 2021a], which introduced a Markov chain-based framework for discrete noise injection and denoising trained via ELBO maximization. This was extended to continuous time by CTMC [Campbell et al., 2022]. In parallel, SEDD [Lou et al., 2023] learned the reverse process by modeling the ratio of marginal probabilities using a denoising score entropy objective, while Masked Diffusion Models such as MDLM [Shi et al., 2024, Sahoo et al., 2024, Zheng et al., 2024] and RADD Ou et al. [2025] provided further theoretical simplifications and formalized connections between parameterizations. A key breakthrough has been the incorporation of diffusion into existing LLM architectures: Diffusion-NAT [Zhou et al., 2023] aligned the denoising process with non-autoregressive decoding, enabling high-speed generation, while models like LLaDA [Nie et al., 2025], DiffuLLaMA [Gong et al., 2025], and Dream [Ye et al., 2025] successfully scaled diffusion-based decoding to billion-parameter models, significantly improving inference efficiency without compromising output quality.

### 2.2 ACCELERATE DIFFUSION-BASED LARGE LANGUAGE MODELS

Followed by mature diffusion large language models, their acceleration methods are also under development. Concretely, dllm-Cache [Liu et al., 2025] proposes a training-free, adaptive caching framework that performs long-interval prompt caching and short-interval, value-similarity–guided partial response updates. Fast-dLLM [Wu et al., 2025] introduces block-wise approximate KV caching and a confidence-aware parallel decoding rule that only decodes tokens whose marginal confidence exceeds a threshold. Hu et al. [2025] propose FreeCache to approximate KV states by reusing stable prompt/block activations across steps. They also introduce Guided Diffusion to decide which tokens to unmask each step without retraining. SlowFast-Sampling [Wei et al., 2025] proposes a dynamic two-stage sampler that alternates a cautious exploratory phase with a fast phase that aggressively decodes high-confidence tokens within that span. Prophet [Li et al., 2025] monitors the top-2 logit gap and commits all remaining tokens in one shot via early-commit decoding once it is sufficiently confident. However, these accelerate methods are often static and lack flexibility. To address this, we propose **Learn2PD**, a novel dynamic remasking method that achieves more efficient inference acceleration by reducing the unnecessary and repetitive decoding steps. Moreover, we also introduce **EoTP** to avoid redundant decoding when the answer does not span the full generation length.

## 3 METHODOLOGY

In this section, we present Learn2PD, a learned approach to accelerate diffusion language model inference through adaptive parallel decoding. We begin by reviewing the fundamentals of diffusion language models and their current parallel decoding strategies (Section 3.1.1). Through empirical analysis, we reveal a critical inefficiency: existing methods unnecessarily remask a significant proportion of correctly predicted tokens, leading to redundant computation (Section 3.1.2). This observation motivates our core contribution—training a lightweight filter model to predict token stability and approximate an oracle parallel decoding strategy (Section 3.2). Finally, we introduce an early-stopping mechanism to further eliminate padding token overhead (Section 3.3).

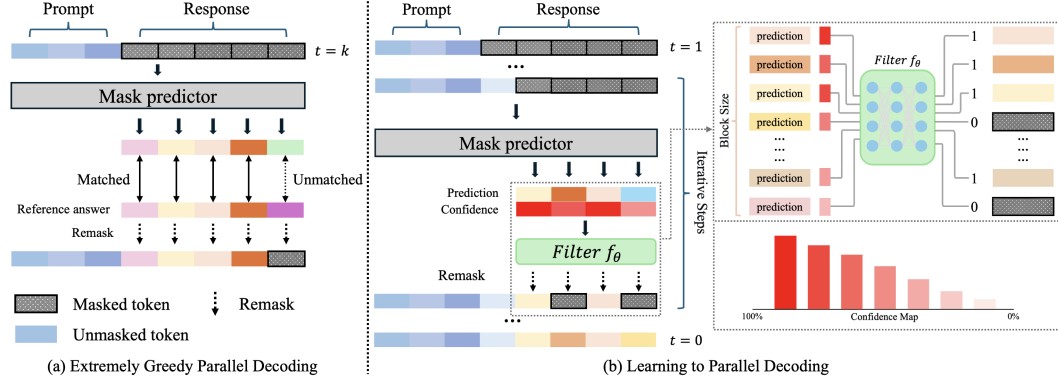

Figure 2: **A Conceptual Overview of pipeline and method.** (a) Extremely Greedy Parallel (EGP). This strategy compares the predicted tokens with the reference answer and only remasks the tokens that do not match in these comparisons. (b) Learning to Parallel Decoding (Learn2PD). During the inference process, after the model generates predictions and confidences for each token, the confidence of each token is fed into a filter model $f_\theta$ to determine which tokens need to be remasked. This determination then guides the subsequent remasking procedure.

## 3.1 PRELIMINARY

### 3.1.1 DIFFUSION LARGE LANGUAGE MODELS

**Forward Process.** Given an input sentence $\boldsymbol{x}_0 \in \{0, 1, \dots, V-1\}^L$ and a noise level $t \in [0, 1]$, where $V$ and $L$ represent the vocabulary size and sentence length. The forward process randomly and independently masks out tokens through the following Markov chain:

$$q_{t|0}(\boldsymbol{x}_t \mid \boldsymbol{x}_0) = \prod_{i=0}^{L-1} \left[ (1-t)\,\mathbf{1}\{\boldsymbol{x}_t^i = \boldsymbol{x}_0^i\} + t \cdot \mathbf{1}\{\boldsymbol{x}_t^i = m\} \right] \tag{1}$$

where $x^i$ denotes the $i$-th element of $\boldsymbol{x}$, $m$ denotes the mask token [Devlin et al., 2019], $\boldsymbol{x}_t$ denotes the noisy data at time $t$, and $q_0(\cdot)$ is the data distribution $p_{\text{data}}(\cdot)$.

**Reverse process.** The reverse process iteratively recovers masked tokens by predicting data distribution from a masked sequence. Transitioning from corruption level $t$ to an earlier level $s$, where $0 \le s < t \le 1$ can be approximated as

$$q_{s|t}(\boldsymbol{x}_s \mid \boldsymbol{x}_t) = \prod_{i=0}^{L-1} q_{s|t}(\boldsymbol{x}_s^i \mid \boldsymbol{x}_t), \quad q_{s|t}(\boldsymbol{x}_s^i \mid \boldsymbol{x}_t) = \begin{cases} 1, & \boldsymbol{x}_t^i \neq m,\ \boldsymbol{x}_s^i = \boldsymbol{x}_t^i, \\ \dfrac{s}{t}, & \boldsymbol{x}_t^i = m,\ \boldsymbol{x}_s^i = m, \\ \dfrac{t-s}{t}\, q_{0|t}(\boldsymbol{x}_s^i \mid \boldsymbol{x}_t), & \boldsymbol{x}_t^i = m,\ \boldsymbol{x}_s^i \neq m, \end{cases} \tag{2}$$

where $m$ represent the [MASK] and $q_{0|t}(\cdot)$ is the data prediction distribution by the model [Ho et al., 2020]. Given a prompt $\boldsymbol{c} = (c_1, ..., c_M)$, the response $y$ is generated in $K$ discrete steps. In each step $k$, a mask predictor $p_\theta$ takes $\mathbf{y}^{(k)}$ as input and predicts the distribution of sequence. The estimate of the sequence $\hat{\boldsymbol{y}}^{(0)}$ is generated via greedy decoding:

$$\hat{\boldsymbol{y}}^{(0)} = \arg\max_{\boldsymbol{y} \in \mathcal{T}} P_\theta\big(\boldsymbol{y} \mid \mathbf{c}, \mathbf{y}^{(k)}\big) = \arg\max_{\boldsymbol{y} \in \mathcal{T}} p_\theta\big(\mathbf{c}, \mathbf{y}^{(k)}; \theta\big) \tag{3}$$

**Low-Confidence Remasking.** To improve the sample quality, the unmasking tokens with low confidence would be remasked. This approach follows a common practice in non-autoregressive generation for improving output fidelity [Ghazvininejad et al., 2019]. For each position $i$, the model predicts $\hat{y_0}^{(k)}$ and computes its confidence $c_i$, which is given by:

$$c_i = P_\theta\big(\hat{y_{0,i}}^{(k)} \mid \mathbf{c}, \mathbf{y}^{(k)}\big) \tag{4}$$

The tokens corresponding to the $n$ lowest confidence would be set to [MASK] again, where $n$ is calculated by the noise level $t$.

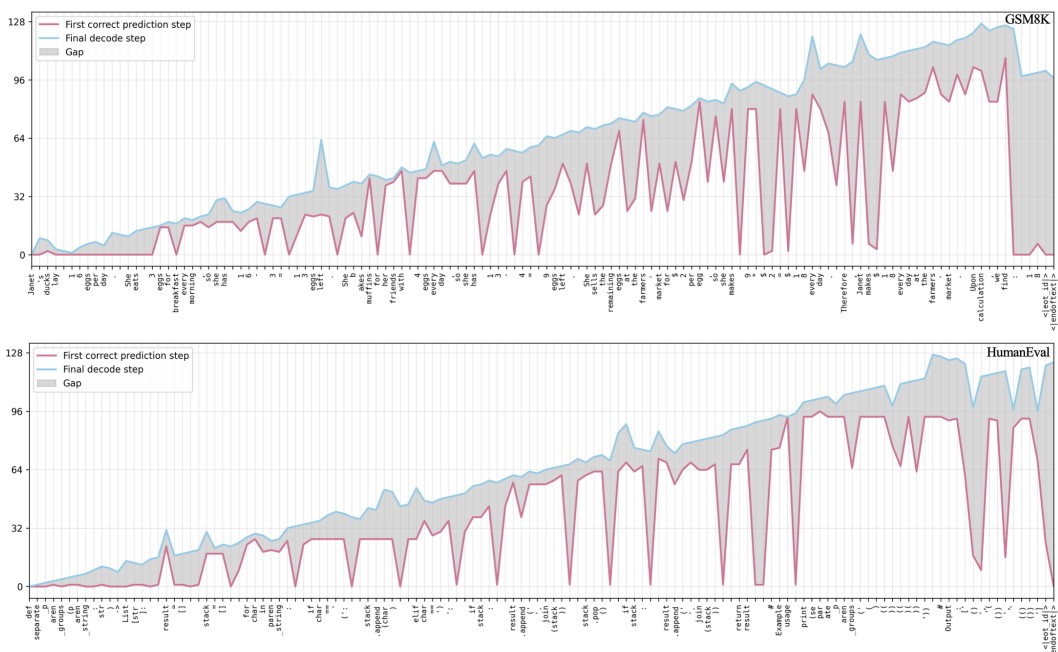

Figure 3: **Samples of gaps in different datasets: GSM8K and HumanEval.** The red line means the first correct prediction step, and the blue line means the actual decoding step.

### 3.1.2 UNNECESSARY REPETITIVE DECODING

Building on the iterative inference process displayed in Section 3.1.1, we investigate the unnecessary and repetitive decoding conditions in diffusion-based large language models. We conducted experimental analyses with LLaDA-8B-Instruct[Nie et al., 2025] on two widely used datasets: GSM8K [Cobbe et al., 2021] and HumanEval [Chen et al., 2021]. We choose LLaDA as our base model due to its state-of-the-art performance and availability of pre-trained checkpoints across multiple scales. During the inference process, the model generates predictions for each masked token and re-masks those with low confidence. To maximize accuracy, existing dLLMs often adopt a conservative re-masking strategy, which results in many correctly predicted tokens being unnecessarily re-masked and re-predicted. Specifically, we measured the extent of redundant and repetitive decoding, defined as the number of times the model continues to decode a token after it has already matched the reference answer. In this paper, we refer to the answer produced by LLaDA under the standard generation process as the **reference answer**.[1]

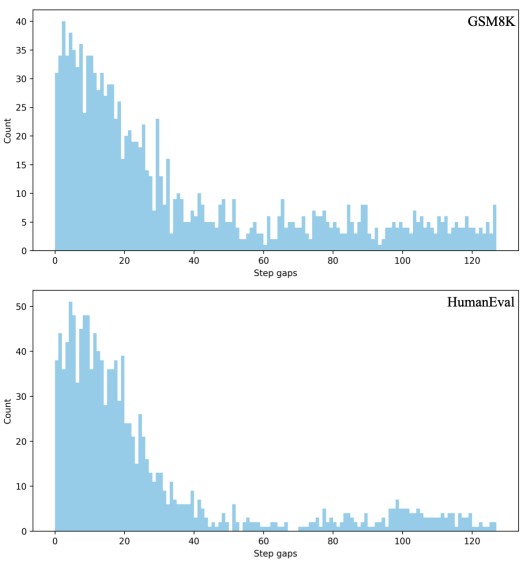

Figure 4: **Distributions of gaps in different datasets: GSM8K and HumanEval.** These two histograms show the distribution of step gaps for each token between the actual decoding step and the step with the first correct prediction.

**Analysis of unnecessary repetition.** As illustrated in Figure 4, we tally up the distribution of the step gaps between the decoding step and the step with the first correct prediction. For each dataset,

---

[1]For all analyses in this section, we set LLaDA's Generation Length at 128 and Block Size at 32.

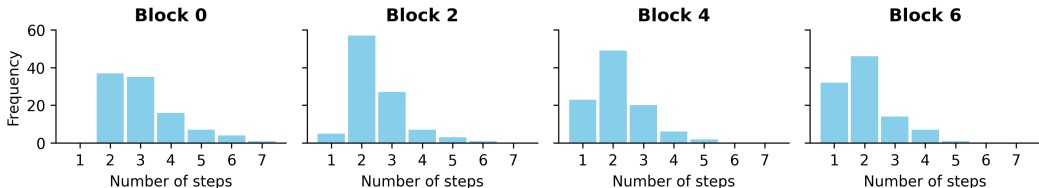

Figure 5: **Distribution of decoding steps per block with Extremely Greedy Parallel (EGP) strategy.** Histograms illustrate the number of decoding steps performed in each block when using our strategy with LLaDA-8B-Instruct on GSM8K based on 100 samples.

we randomly sample 10 questions to conduct the experiment. We find that most of the tokens still need to be decoded more than 10 times, even though they are already correct. And Figure 3 shows one sample from each dataset. The red line means the first correct prediction step, and the blue line means the actual decoding step. It is clear that the model performs many unnecessary decoding steps before unmasking the tokens.

### 3.1.3 EXPECTED INFERENCE PROCESS: EXTREMELY GREEDY PARALLEL (EGP)

Based on the above findings, we observe that a large portion of tokens are remasked as `[MASK]` and decoded multiple times even after they have already been decoded to the reference answer. Motivated by this, we define the **Extremely Greedy Parallel (EGP)** oracle (Figure 2a) as: at each step k, unmask token $i$ if and only if $M(x^k)_i = y_i$, where $y_i$ is the reference answer for token $i$. This oracle achieves optimal speedup by never remasking correct predictions.

**Acceleration Potential.** To evaluate the efficiency of our strategy, we compared the number of decoding steps required per block for LLaDA-8B-Instruct on the GSM8K dataset under the **Extremely Greedy Parallel** policy versus the standard decoding regime. Similarly, we fix the Generation Length to 256 and the Block Size to 32.

As shown in Figure 5, the results are striking. Our strategy achieves a median of 2 decodings per block while maintaining the same accuracy. In contrast, LLaDA with the vanilla setting requires 32 decodings per block. This demonstrates a substantial opportunity for efficiency gains, without compromising output quality.

### 3.2 LEARNING TO PARALLEL DECODING

Although our Extremely Greedy Parallel strategy performs well, this oracle requires ground truth tokens that are unavailable during inference. To address this, we propose a novel approach: **Learning to Parallel Decoding (Learn2PD)** (Figure 2b). Our goal is to simulate the EGP strategy after each decoding step to select tokens and decide whether to remask. We can reformulate this as an optimization problem by using Binary Cross-Entropy Loss (BCELoss) [De Boer et al., 2005]:

$$\arg\min \ -\frac{1}{m}\sum_{i=1}^{m}\Big[y_i\log p_i + (1-y_i)\log(1-p_i)\Big] \tag{5}$$

where $y_i$ indicates whether token $t_i$ should be remasked under the EGP strategy: 0 means it should be remasked and 1 means it can be unmasked. During the inference process, a threshold $\tau$ is applied to discretize $p_i$ into either 0 or 1. The $p_i$ is generated from which we called the filter model $f_\theta$. In this algorithm, the only trained parameters are $\theta$. Therefore, the parameters of the diffusion large language model remain unchanged. Then, the training loss should be:

$$\mathcal{L}_{BCE} = -\frac{1}{m}\sum_{i=1}^{m}\Big[y_i\log\sigma(z_i) + (1-y_i)\log(1-\sigma(z_i))\Big] \tag{6}$$

where $z_i$ is the output of the filter model (logit). We show the algorithms for training and inference in Algorithm 1 and Algorithm 2. The filter $f_\theta$ takes the confidence of the prediction as input and returns the logits $z$ to indicate the probability of no remask. And in order to ensure $z$ remains in

the range $[0, 1]$, we apply a sigmoid function on $z$ before it is passed to the dLLMs. Critically, the filter model $f_\theta$ adds negligible overhead during inference. Our experimental results in Section 4 quantitatively demonstrate that the achieved speedup vastly outweighs this minimal overhead.

---

**Algorithm 1** Training

**Require:** Diffusion large language model $M$, filter model $f_\theta$, prompt set $x_{\text{prompt}}$, reference answer set $x_{\text{reference}}$, generation length $L_{\text{gen}}$, learning rate $\eta$, block size $s$
1: **repeat**
2:     $x_i \in x_{\text{prompt}}, r_i \in x_{\text{reference}}, l_i = length(x_i)$
3:     $X \leftarrow \text{concat}(x_i, [\text{MASK}]^{L_{\text{gen}}})$
4:     **for** $b = 0, ..., \frac{L_{\text{gen}}}{s} - 1$ **do**
5:         $\mathcal{M} \leftarrow \{1, 2, ..., s\}$
6:         **while** $\mathcal{M} \neq \emptyset$ **do**
7:             $conf_t, pre_t = M(X)$
8:             **if** $pre_{t,j} = r_{i,j}$ **then**
9:                 $\hat{y_j} \leftarrow 1, \mathcal{M} \leftarrow \mathcal{M} \setminus \{j\}$
10:                 $X_{l_i+b \cdot s+j} \leftarrow pre_{t,j}$
11:             **else**
12:                 $\hat{y_j} \leftarrow 0$
13:             **end if**
14:             $\mathcal{L} \leftarrow \text{BCELoss}(\hat{y}, f_\theta(conf_t))$
15:             $\theta \leftarrow \theta - \eta \cdot \nabla_\theta L$
16:         **end while**
17:     **end for**
18: **until** converged

---

**Algorithm 2** Inference

**Require:** Diffusion large language model $M$, filter model $f_\theta$, prompt set $x_{\text{prompt}}$, generation length $L_{\text{gen}}$, block size $s$, filter threshold $\tau$
1: **for** each $x_i \in x_{\text{prompt}}$ **do**
2:     $l_i = length(x_i), X \leftarrow \text{concat}(x_i, [\text{MASK}]^{L_{\text{gen}}})$
3:     **for** $b = 0, ..., \frac{L_{\text{gen}}}{s} - 1$ **do**
4:         $\mathcal{M} \leftarrow \{1, 2, ..., s\}$
5:         **while** $\mathcal{M} \neq \emptyset$ **do**
6:             $conf_t, pre_t = M(X), logit_t = f_\theta(conf_t)$
7:             **if** $logit_{t,j} > \tau$ **then**
8:                 $\mathcal{M} \leftarrow \mathcal{M} \setminus \{j\}, X_{i+b \cdot s+j} \leftarrow pre_{t,j}$
9:             **end if**
10:         **end while**
11:     **end for**
12:     $\text{response}_i = X_{l_i:l_i+L_{\text{gen}}-1}$
13: **end for**
14: **return** response

---

### 3.3 END-OF-TEXT PREDICTION

Besides the methods mentioned earlier, we observed that when the generation length of a diffusion large language model is increased to 1024, the generation time rises significantly for the same question compared to a length of 256, even though the final answer length remains unchanged. According to the analysis of the generated output, we find that the extra length is filled with the [EoT] token, and the additional decoding time is spent repeatedly decoding the [EoT] token. Based on this, we propose the **End-of-Text Prediction (EoTP)** approach: at each step, when a [EoT] token is unmasked, discard tokens in subsequent positions. The shortened sequence is then used as the input of next step. Therefore, we update the inference process to handle the long-generation-length challenge in Appendix B. We illustrate the details in Figure 6.

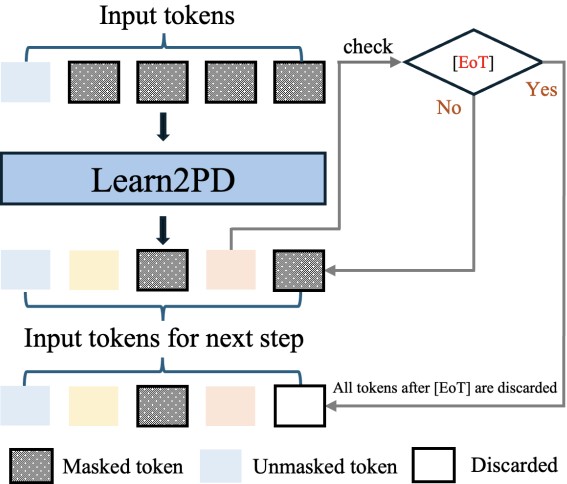

Figure 6: **Schematic of the End-of-Text Prediction Policy.** During the inference process, upon detection of an [EoT] token in a decoded block, all subsequent tokens are discarded.

Our analysis shows that 89.59% of computational cost comes from decoding padding tokens after [EoT]. EoTP yields substantial computational savings by dynamically reducing the effective input length throughout the diffusion process. The experiments and relevant analysis are in Appendix D.

## 4 EXPERIMENT

### 4.1 EXPERIMENTAL SETTINGS

**Models and Datasets.** We implement our methods on the representative dLLM: LLaDA-8B-Instruct [Nie et al., 2025] to measure the acceleration of the inference process across various benchmarks. To ensure the broad applicability of the methods, we conducted experiments on four datasets covering three different types of problems, which are GSM8K[Cobbe et al., 2021], Math

Table 1: Benchmark results on the LLaDA-8B-Instruct suite. Each method was evaluated using two generation lengths (256 and 1024) across four datasets. Performance is measured using three metrics: TPS (tokens/sec), speedup, and accuracy score. The highest throughput and speedup values for each configuration are highlighted in bold.

| Task | Methods | Gen Length | Inference Efficiency | | Performance |
|---|---|---|---|---|---|
| | | | TPS↑ | Speed (TPS)↑ | Score |
| GSM8K (5-shot) | LLaDA-8B-Instruct | 256 | 3.41 | 1.00× | 78.70 |
| | | 1024 | 0.54 | 1.00× | 77.60 |
| | + Learn2PD | 256 | $14.07_{+10.66}$ | 4.13× | 78.62 |
| | | 1024 | $6.63_{+6.09}$ | 12.21× | 77.26 |
| | Learn2PD + EoTP | 256 | $14.35_{+10.94}$ | 4.21× | 78.62 |
| | | 1024 | $12.26_{+11.72}$ | **22.58×** | 79.83 |
| Math (4-shot) | LLaDA-8B-Instruct | 256 | 4.70 | 1.00× | 32.90 |
| | | 1024 | 1.70 | 1.00× | 35.21 |
| | + Learn2PD | 256 | $15.16_{+10.46}$ | 3.21× | 32.22 |
| | | 1024 | $10.98_{+9.28}$ | 6.45× | 34.01 |
| | Learn2PD + EoTP | 256 | $15.21_{+10.51}$ | 3.23× | 31.40 |
| | | 1024 | $12.27_{+10.57}$ | **7.22×** | 34.60 |
| HumanEval (0-shot) | LLaDA-8B-Instruct | 256 | 3.33 | 1.00× | 39.63 |
| | | 1024 | 0.53 | 1.00× | 37.21 |
| | + Learn2PD | 256 | $11.66_{+8.33}$ | 3.5× | 38.41 |
| | | 1024 | $4.63_{+4.10}$ | 8.78× | 37.84 |
| | Learn2PD + EoTP | 256 | $11.88_{+8.55}$ | 3.57× | 38.41 |
| | | 1024 | $6.63_{+6.10}$ | **12.55×** | 35.98 |
| MBPP (3-shot) | LLaDA-8B-Instruct | 256 | 3.14 | 1.00× | 31.22 |
| | | 1024 | 0.58 | 1.00× | 10.61 |
| | + Learn2PD | 256 | $14.96_{+11.82}$ | 4.77× | 30.84 |
| | | 1024 | $6.96_{+6.38}$ | 12.08× | 10.04 |
| | Learn2PD + EoTP | 256 | $15.88_{+12.74}$ | 5.06× | 31.03 |
| | | 1024 | $9.89_{+9.31}$ | **17.16×** | 11.02 |

[Lewkowycz et al., 2022], HumanEval[Chen et al., 2021], and MBPP [Austin et al., 2021b]. All experiments are conducted on 4 NVIDIA A6000 GPUs.

**Filter Model $f_\theta$ Training.**    To train a filter model that can be applied to a wide range of tasks, we selected 40 samples from each of the 66 types of questions in the FLAN dataset, resulting in a total of 2,640 samples for training. In this experiment, we used the simplest two-layer MLP as our filter model. Since the dLLMs remains frozen and only $f_\theta$ is trained, the number of trainable parameters is extremely limited. For example, for an LLaDA with a block size of 32, the total number of trainable parameters is only 2,112. We trained $f_\theta$ for 5,000 epochs until the model converged. The learning rate is set to 0.001, and the AdamW optimizer is used to optimize $f_\theta$.

Our training process consists of two stages. In the first stage, samples are collected by following an Extremely Greedy Parallel policy, recording the confidence scores and token selections at each step during parallel decoding. This data is then used in the second stage to train a filter model $f_\theta$. The data collection in the first stage was conducted on 4 NVIDIA RTX A6000 GPUs and took approximately three hours. The subsequent training of the filter model in the second stage was deployed on a T4 GPU and required only 6 minutes. The details of training are in Appendix E.

**Evaluation.**    We evaluate the inference acceleration and generation quality of **Learn2PD** and **EoTP** methods by using quantitive metrics. The inference speed is quantified with Tokens Per Second (TPS), indicating the average number of tokens generated per second. And the generation quality is measured in task-specific metrics, such as accuracy for GSM8K, showing the model's

performance with acceleration methods. In addition to this, we set the Generation Length to 256 & 1024 and the Block Size to 32.

## 4.2 MAIN RESULTS

We present the inference performance and efficiency profits for Learn2PD and EoTP on the LLaDA-8B-Instruct across four benchmarks, as shown in Table 1.

In summary, Learn2PD significantly enhances inference efficiency across all tasks. Compared to the baseline model, our optimal method typically achieves a 3 to 4 times speedup at a generation length of 256 and a 6 to 12 times speedup at a generation length of 1024. When EoTP is incorporated, the improvements become even more pronounced, particularly with a generation length of 1024. For instance, combining Learn2PD and EoTP results in a throughput increase of 22.58× (on GSM8K, 5-shot) and 17.16× (on MBPP, 3-shot) relative to the baseline. These results demonstrate that our methods are not only effective individually but also highly orthogonal, resulting in compounded acceleration. More importantly, these efficiency gains have negligible impact on accuracy. The performance scores of our accelerated methods remain within 1–2 points of the baseline, and in some cases, the score is even slightly improved.

## 4.3 COMPATIBILITY WITH KEY-VALUE CACHE

We further evaluate the compatibility of our approach with established Key-Value (KV) Cache techniques by integrating both Dual Cache and Prefix Cache strategies [Wu et al., 2025]. Experiments are conducted on GSM8K with a generation length of 1024 tokens. As summarized in Table 2, the baseline model (Learn2PD & EoTP) achieves a throughput of 12.26 TPS, a speed-up of 22.58×, and an accuracy score of 79.83. When augmented with the Dual Cache, the system attains substantially higher efficiency, reaching 31.23 TPS and a 57.51× speedup, albeit with a slight decrease in accuracy (74.00). Similarly, incorporating the Prefix Cache also brings noticeable improvements, yielding 14.79 TPS and a 27.23× acceleration while maintaining a competitive score of 77.71. These results confirm that our method is orthogonal to and fully compatible with standard KV caching mechanisms, demonstrating its ability to leverage such strategies to enhance inference efficiency.

Table 2: A comparison of our method with and without KV Cache. The results show a significant performance improvement when augmented with both Dual and Prefix Caches, underscoring that our method is orthogonal to and fully compatible with existing KV caching strategies.

Table 3: A comparison of the acceleration performance using filter models of varying complexity (represented by the number of MLP layers). The results indicate that a two-layer MLP model achieves the optimal balance by providing significant speedup.

| Methods | TPS | Speed | Score |
|---|---|---|---|
| Learn2PD & EoTP | 12.26 | 22.58× | 79.83 |
| + Dual Cache | 31.23 | 57.51× | 74.00 |
| + Prefix Cache | 14.79 | 27.23× | 77.71 |

| # Layers | TPS | Speed | Score |
|---|---|---|---|
| Single-layer | 8.77 | 2.57× | 78.62 |
| Two-layer | 14.07 | 4.13× | 78.62 |
| Four-layer | 11.41 | 3.35× | 78.85 |

## 4.4 ANALYSIS: ABLATION STUDY

**Effect of Filter Model $f_\theta$ Complexity.** To investigate the impact of the filter model's architectural complexity on acceleration performance, we conduct an ablation study using MLP-based filter models with varying depths. As illustrated in Table 3, a two-layer MLP achieves the highest throughput (14.07 TPS) and speed-up (4.13×), while maintaining 78.62% accuracy. In comparison, the single-layer model yields lower efficiency with a similar accuracy, suggesting limited representational capacity. Although the four-layer model attains a marginally better accuracy score, it results in reduced inference speed, indicating increased computational overhead. These results demonstrate that a two-layer configuration offers the optimal trade-off between efficiency and predictive performance, effectively balancing model complexity and acceleration gain.

**Effect of Generation Length.** To examine the impact of generation length on the performance of our methods, we compare the speedup and accuracy of Learn2PD and its enhanced variant (Learn2PD & EoTP) across varying output lengths. As shown in Table 4, both methods achieve greater acceleration as the generation length increases, while consistently maintaining competitive scores. At a shorter length of 128, Learn2PD & EoTP reaches a speed-up of 3.36×. And the speed-up improves steadily with longer sequences, culminating in a substantial 22.58× acceleration at a length of 1024. These results indicate that our approach is particularly effective for long-sequence generation, efficiently reducing the unnecessary decodings to maximize inference speed without compromising output quality.

Table 4: Performance comparison of our methods across different generation lengths. While maintaining a comparable accuracy, both Learn2PD and EoTP deliver substantially greater speedup at a length of 1024 compared to shorter sequences.

| Gen Length | Methods | Speed | Score |
|---|---|---|---|
| 128 | Learn2PD | 3.29× | 73.92 |
| | Learn2PD & EoTP | 3.36× | 74.07 |
| 256 | Learn2PD | 4.13× | 78.62 |
| | Learn2PD & EoTP | 4.21× | 78.62 |
| 512 | Learn2PD | 6.66× | 77.71 |
| | Learn2PD & EoTP | 7.60× | 79.68 |
| 1024 | Learn2PD | 12.21× | 77.26 |
| | Learn2PD & EoTP | 22.58× | 79.83 |

**Effect of Filter Model $f_\theta$ threshold $\tau$.** We perform an ablation study to examine the impact of the filtering threshold on inference accuracy and throughput. As shown in Figure 7, reducing the threshold improves throughput but leads to a corresponding decline in accuracy. For example, at $\tau = 0.99$, the model achieves a throughput of 4.68 TPS (vs. baseline 3.41 TPS) with 78.92% accuracy. In contrast, lowering the threshold to $\tau = 0.9$ causes a more pronounced reduction in accuracy. The results indicate that a threshold of $\tau = 0.96$ offers an optimal balance, delivering both high throughput (4.13× speedup) and near-baseline accuracy. These findings underscore the critical role of the filtering threshold in achieving an effective trade-off between inference efficiency and output quality.

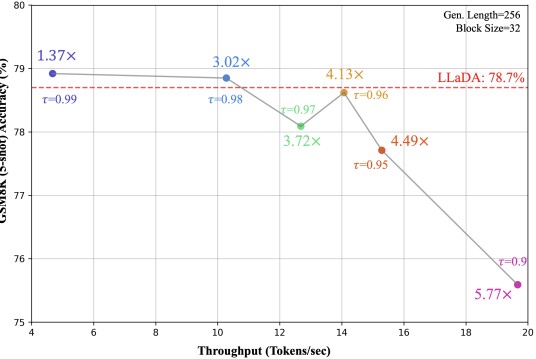

Figure 7: **Impact of the filtering threshold on accuracy and throughout.** We find that a threshold of 0.96 represents a favorable balance, maintaining high accuracy and comparable inference speed.

## 5 CONCLUSION

In this work, we investigate the issue of extensive repetitive decoding during inference in Diffusion-based Large Language Models. To enable timely unmasking of correctly predicted tokens, we propose **Learn2PD**, a parallel decoding architecture that employs a filter model to make case-specific selections. This filter model is lightweight and pre-trained, thus requiring no additional training during inference. Furthermore, to address the time overhead caused by repeated encoding of the [EoT] token as the generation length increases, we introduce the **EoTP** mechanism, which halts decoding immediately after [EoT] is generated, thereby reducing unnecessary computational cost. Extensive experiments across multiple benchmarks and model baselines (LLaDA) demonstrate that our approach achieves up to **22.58×** speedup without sacrificing accuracy—and up to **57.51×** when combined with KV Cache. Our proposed method offers a compelling solution for deploying diffusion-based LLMs as alternatives to autoregressive models in future applications.

## ETHICS STATEMENT

Potential for Misuse: We acknowledge that our acceleration techniques, by lowering the computational cost of inference, could inadvertently lower the barrier for malicious use. This could enable bad actors to scale up harmful applications such as disinformation campaigns, spam, phishing, or automated malicious code generation more efficiently.

## REPRODUCIBILITY STATEMENT

To support reproducibility, a complete anonymized code repository is provided as supplementary material. The repository encompasses all necessary components to replicate our work: the implementation source code for the proposed model and algorithms, the scripts required to run the experiments, comprehensive hyperparameter configuration files, and detailed execution instructions.

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

## A    THE USE OF LARGE LANGUAGE MODELS

Large language models (LLMs) were used in this work exclusively for the purpose of text polishing and refinement. Their role was strictly limited to assisting with grammatical correction, improving sentence fluency, and enhancing word choice to increase the overall clarity and readability of the manuscript.

## B    UPDATE INFERENCE ALGORITHM

---

**Algorithm 3 Update Inference**

---

**Require:** Diffusion large language model $M$, filter model $f_\theta$, prompt set $x_{\text{prompt}}$, generation length $L_{\text{gen}}$, block size $s$, filter threshold $\tau$
1: **for** each $x_i \in x_{\text{prompt}}$ **do**
2:      $l_i \leftarrow length(x_i)$, $X \leftarrow \text{concat}(x_i, [\text{MASK}]^{L_{\text{gen}}})$
3:      **for** $b = 0, ..., \frac{L_{\text{gen}}}{s} - 1$ **do**
4:          $\mathcal{M} \leftarrow \{1, 2, ..., s\}$
5:          **while** $\mathcal{M} \neq \emptyset$ **do**
6:              $conf_t, pre_t = M(X)$, $logit_t = f_\theta(conf_t)$
7:              **if** $logit_{t,j} > \tau$ **then**
8:                  $\mathcal{M} \leftarrow \mathcal{M} \setminus \{j\}$, $X_{i+b\cdot s+j} \leftarrow pre_{t,j}$
9:              **end if**
10:          **end while**
11:          **if** [endoftext] in $X$ **then**
12:              **break**
13:          **end if**
14:      **end for**
15:      $response_i = X_{l_i:l_i+L_{\text{gen}}-1}$
16: **end for**
17: **return** $response$

---

## C    EXPERIMENTS ON DREAM MODEL

Table 5: Results of Learn2PD and EoTP mechanisms on Dream in different generation lengths.

| Generation Length | Methods | TPS | Speed | Score |
|---|---|---|---|---|
| 256 | Dream | 2.38 | 1.00× | 74.6 |
| | + Learn2PD | 4.04 | 1.70× | 74.5 |
| | Learn2PD & EoTP | 4.73 | 1.99× | 73.4 |
| 1024 | Dream | 0.38 | 1.00× | 71.9 |
| | + Learn2PD | 2.19 | 5.84× | 72.1 |
| | Learn2PD & EoTP | 4.12 | 10.99× | 73.1 |

The table presents the results of the Learn2PD and EoTP mechanisms applied to the Dream model across two generation lengths: 256 and 1024. For a generation length of 256, adding Learn2PD increased the TPS to 4.04 and speed to 1.70×, with a slightly higher score of 74.5. Combining Learn2PD and EoTP further improved TPS to 4.73 and speed to 1.99×. For a generation length of 1024, incorporating Learn2PD significantly boosted TPS to 2.19 and speed to 5.84×, with a marginally higher score of 72.1. Combining Learn2PD and EoTP yielded the highest TPS of 4.12 and speed of 10.99×. These results demonstrate that Learn2PD and EoTP could considerably accelerate the Dream model with only a minimal loss in performance.

# D    EXPERIMENTS AND ANALYSIS ON EOTP MECHANISM

Table 6: A comparison of LLaDA and EoTP mechanisms in different generation lengths.

| Methods | Generation Length | TPS | Speed | Score |
|---|---|---|---|---|
| LLaDA | 256 | 3.41 | 1.00× | 78.70 |
| | 512 | 1.67 | 1.00× | 77.71 |
| | 1024 | 0.54 | 1.00× | 77.62 |
| + EoTP | 256 | 3.76 | 1.10× | 79.23 |
| | 512 | 3.38 | 2.02× | 78.77 |
| | 1024 | 3.25 | 5.98× | 79.38 |

This table 6 clearly demonstrates the significant advantage of integrating the EoTP mechanism with the base LLaDA model, particularly for long-sequence generation. While the standalone LLaDA model exhibits substantial performance degradation as generation length increases—evidenced by the sharp decline in TPS from 3.41 to 0.54—the incorporation of EoTP not only mitigates this degradation but also delivers considerable speedup. Most notably, at a sequence length of 1024, EoTP achieves a dramatic 5.98× acceleration while simultaneously improving output quality, with the Score increasing from 77.62 to 79.38. These gains can be largely attributed to the elimination of redundant computation: quantitative analysis shows that 89.59% of the baseline computational cost arises from decoding padding tokens after the `[EoT]` token. EoTP effectively removes this overhead by dynamically detecting sequence completion once all original non-`[MASK]` positions are unmasked, thereby optimizing inference efficiency without compromising performance.

# E  TRAINING CURVE OF FILTER MODEL

As shown in Figure 8, this learning curve illustrates that both training (blue) and validation (red) loss fall sharply in the earliest epochs (from 0.68 to 0.28), then decrease much more gradually over the full 5,000-epoch run, approaching a plateau near 0.21–0.23. The validation curve remains slightly above the training curve throughout, indicating only a modest generalization gap rather than pronounced overfitting; the parallel, steady decline shows the model continues to improve on unseen data but with diminishing returns. The long, flat tail of both curves indicates the model has effectively converged.

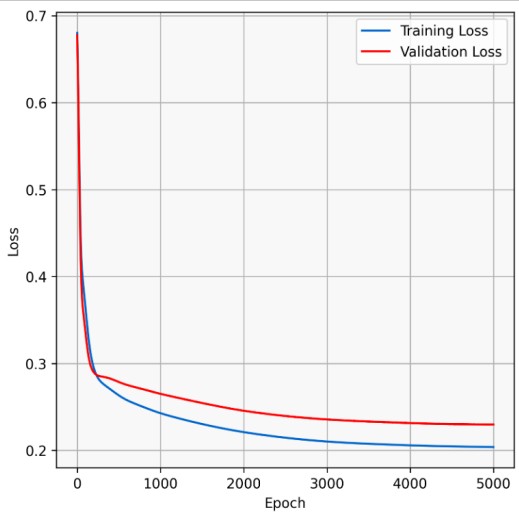

Figure 8: **Learning curve of filter model** $f_\theta$. The learning curves illustrate the progression of training and validation loss across 5,000 epochs, with the former in blue and the latter in red.

