# OpenReview forum: "Learning to Parallel: Accelerating Diffusion Large Language Models via Learnable Parallel Decoding"
_ICLR.cc/2026/Conference — ICLR 2026 Poster_

### Official Review · Reviewer_vfNL · 2025-10-14

**Soundness:** 3
**Presentation:** 2
**Contribution:** 3
**Rating:** 6
**Confidence:** 3

**Summary:**

The paper proposes Learn2PD, a learnable parallel decoding framework for dLLMs that replaces fixed heuristics with a lightweight filter model that predicts, per token, whether its current prediction already matches the final output.
This adaptive filter enables tokens to be “unmasked” dynamically and in parallel, improving the speed–quality tradeoff. They also introduce an EoTP mechanism to detect completion and prevent redundant decoding, and experiments show significant speedups.

**Strengths:**

1. The problem this paper tries to address is an important one: current dLLM decoding is mainly heuristic-based. A hyperparameter needs to be used (like confidence) for unmasking. The neural predictor is indeed a plausible solution.

2. The speedup achieved is impressive.

3. The evaluation is qutie solid, including compatibility with KV cache methods.

**Weaknesses:**

1. Using a KV cache during inference can cause a misalignment with the training process, potentially altering the model's generated outputs from the training ground truth.

2. As shown in Table 2, there is a big acc drop with dual cache (5.83). Is this caused by Learn2PD?

3. The evaluation is mainly on LLaDA. How does the method generalize to other models like Dream?

4. How much is the inference overhead is the additional predictor?

5. How does the performance change with respect to the size of the predictor?

**Questions:**

Please see the weaknesses above.

---

> ### Author Response · Authors · 2025-11-24
> **Response to Reviewer vfNL**
>
> We thank reviewer vfNL for the insightful and valuable comments. We sincerely hope that our rebuttal has addressed your concerns. If there are any remaining issues or points that are unclear, we would be grateful to receive further feedback and will gladly continue to respond and improve the work accordingly.
>
> **1. W1: Using a KV cache during inference can cause a misalignment with the training process, potentially altering the model's generated outputs from the training ground truth.**
>
> **Reply:** We appreciate your insightful comment. We would like to emphasize that KV cache is introduced only to demonstrate that our method can coexist with other acceleration methos, rather than as a core component of Learn2PD. Yes. Using a KV cache during inference can change the distribution of confidence scores, which can affect the model’s accuracy. To address this, we retrain the filter model specifically under KV-cache-enabled settings so that it can effectively handle the distributional shift introduced by the cache and better align with the cached inference behavior.
>
> | Gen Length | Methods      | Speed  | Accuracy    |
> |-----------|--------------|--------|---------|
> | 1024      | LLaDA & Parallel Decoding & Dual Cache (Fast-dllm) [1]      | 19.60×  | 0.747  |
> |           | Learn2PD (original) & EoTP & Dual Cache   | 57.51×  | 0.740   |
> |           | Learn2PD (Dual Cache) & EoTP & Dual Cache   | 60.47× | 0.753 |
>
> [1] Wu, Chengyue, et al. "Fast-dLLM: Training-free Acceleration of Diffusion LLM by Enabling KV Cache and Parallel Decoding." arXiv (2025). https://arxiv.org/abs/2505.22618.
>
> **2. W2: As shown in Table 2, there is a big acc drop with dual cache (5.83). Is this caused by Learn2PD?**
>
> **Reply:** We appreciate this question. The accuracy drop is mainly due to the Dual Cache mechanism itself. As shown in the previous table, the accuracy already decreases significantly when Dual Cache is enabled, even without Learn2PD. Our method only introduces a relatively minor additional drop. To address this, we retrain the filter specifically under the Dual Cache setting to better align with its confidence distribution, which recovers the accuracy from 74.0% to 75.3%.
>
> **3. W3: The evaluation is mainly on LLaDA. How does the method generalize to other models like Dream?**
>
> **Reply:**   We appreciate your helpful comment. We have conducted GSM8K experiments on Dream under generation lengths of 256 and 1024. The results show that Learn2PD achieves 1.70× and 5.84× speedups over the vanilla Dream model at generation lengths 256 and 1024, respectively. Furthermore, EoTP improves the speedup to 10.99× at generation length 1024. These results suggest that both Learn2PD and EoTP are not limited to the LLaDA model and can transfer effectively to causal diffusion dLLMs such as Dream. We have updated the manuscript to include these results in Appendix and highlighted the corresponding entries in red.
>
> | Gen Length | Methods       | TPS  | Speed  | Accuracy    |
> |-----------|--------------|-------|--------|---------|
> | 256       | Dream           | 2.38  | 1.00×  | 0.746  |
> |           | + Learn2PD | 4.04 | 1.70×  | 0.745   |
> |           | Learn2PD & EoTP    | 4.73 | 1.99× | 0.734 |
> | 1024       | Dream          | 0.38 | 1.00×  | 0.719  |
> |           | + Learn2PD  | 2.19 | 5.84×  | 0.721   |
> |           | Learn2PD & EoTP    | 4.12 | **10.99×** | 0.731 |
>
> **4. W4: How much is the inference overhead is the additional predictor?**
>
> **Reply:**  We appreciate this comment. The additional predictor has only 2,112 parameters, which is negligible compared to the billions of parameters in the LLaDA. The inference impact caused by the filter model is minimal. Therefore, our measured results reflect the actual acceleration. We have also provided the code in the supplementary materials.
>
> **5. W5: How does the performance change with respect to the size of the predictor?**
>
> **Reply:**  Thanks for this comment. We analyze the impact of the predictor (filter model) size on performance in Section 4.4. Specifically, we found that a two-layer MLP performs better than a single-layer MLP, whereas increasing the depth to three layers hurts performance likely due to overfitting.

---

### Official Review · Reviewer_yNHD · 2025-10-30

**Soundness:** 2
**Presentation:** 2
**Contribution:** 3
**Rating:** 6
**Confidence:** 3

**Summary:**

This paper proposes Learn2PD and EoTP, two methods to accelerate inference in diffusion-based large language models (dLLMs), addressing inefficiencies in current parallel decoding approaches. Learn2PD trains a lightweight and adaptive filter model to predict whether the current prediction matches the final output. EoTP detects decoding completion at the end of the sequence, avoiding redundant decoding of padding tokens. With these methods, they achieve significant speedup without performance drop.

**Strengths:**

Decoding strategy for dllm is a very important topic. Unlike autoregressive models, dllm needs to decide which token to decode and which position to decode in each denoising step, which is much harder. This paper proposes an inspiring method by learning to predict the position that is ready to be decoded.

Experimental results show that this method can accelerate the inference significantly.

The filter model only takes the confidence score as input and exhibits very good prediction results. This means that the logits contain rich information that can be leveraged.

**Weaknesses:**

**Section 3.1.2 is difficult to understand.**
> we measured the amount of unnecessary and repetitive decoding, which is defined as the number of times the model continues to decode a token after that token has first matched the reference answer

What does this sentence mean? Does this mean that the model predicts the correct token but fails to unmask it?

**Figure 2 and 4 are hard to read.**

**Limited Evaluation benchmarks and baselines**

**Questions:**

see weaknesses

---

> ### Author Response · Authors · 2025-11-24
> **Response to Reviewer yNHD**
>
> We thank reviewer yNHD for the insightful and valuable comments. We sincerely hope that our rebuttal has addressed your concerns. If there are any remaining issues or points that are unclear, we would be grateful to receive further feedback and will gladly continue to respond and improve the work accordingly.
>
> **1. W1: Section 3.1.2 is difficult to understand.**
> "We measured the amount of unnecessary and repetitive decoding, which is defined as the number of times the model continues to decode a token after that token has first matched the reference answer."
> **What does this sentence mean? Does this mean that the model predicts the correct token but fails to unmask it?**
>
> **Reply:** We appreciate this comment. In the original setting, the model generates a prediction for every masked token, but it does not “trust” all these predictions. Tokens with low confidence are re-masked and passed back to the model for another step of prediction. To maximize accuracy, existing dLLMs typically use a conservative strategy when deciding which tokens to re-mask, so even many tokens that have already been correctly predicted are re-masked and predicted again. And Section 3.1.2 investigates how many of these already-correct tokens are unnecessarily re-masked and decoded again. We update this paragraph in the revised version and highlight it in red.
>
> **2. W2: Figure 2 and 4 are hard to read.**
>
> **Reply:** Thank you for pointing this out. We have updated Figures 2 and 4 in the revised version to improve their readability.
>
> **3. W3: Limited Evaluation benchmarks and baselines**
>
> **Reply:**  We appreciate your comment. We have conducted GSM8K experiments on Dream under generation lengths of 256 and 1024. The results show that Learn2PD achieves 1.70× and 5.84× speedups over the vanilla Dream model at generation lengths 256 and 1024, respectively. Furthermore, EoTP improves the speedup to 10.99× at generation length 1024. These results suggest that both Learn2PD and EoTP are not limited to the LLaDA model and can transfer effectively to causal diffusion dLLMs such as Dream. We have updated the manuscript to include these results in Appendix and highlighted the corresponding entries in red.
>
> | Gen Length | Methods       | TPS  | Speed  | Accuracy    |
> |-----------|--------------|-------|--------|---------|
> | 256       | Dream           | 2.38  | 1.00×  | 0.746  |
> |           | + Learn2PD | 4.04 | 1.70×  | 0.745   |
> |           | Learn2PD & EoTP    | 4.73 | 1.99× | 0.734 |
> | 1024       | Dream          | 0.38 | 1.00×  | 0.719  |
> |           | + Learn2PD  | 2.19 | 5.84×  | 0.721   |
> |           | Learn2PD & EoTP    | 4.12 | **10.99×** | 0.731 |
>
> In addition, we compare our method against other strategies. Their performance is displayed directly in their papers. We do not re-implement their methods.
>
> | Methods                    | GSM8K  | MATH   | HumanEval | MBPP     |
> |-------------------------|-----------|----------|---------------|-----------|
> | Fast-dLLM-Parallel [1]   | 2.5×       |  2.7×     | 3.3×             | 4.1×       |
> | SlowFast Sampling [2]   | 3.2×       | 2.2×      | 3.2×             | 2.7×       |
> | Learn2PD (ours)        | **4.1×**  | **3.2×** | **3.5×**        | **4.8×**  |
> All methods achieve accuracy scores that closely match those of the original LLaDA.

---

### Official Review · Reviewer_pSk7 · 2025-10-30

**Soundness:** 2
**Presentation:** 3
**Contribution:** 2
**Rating:** 4
**Confidence:** 4

**Summary:**

This paper addresses the autoregressive decoding bottleneck in large language models (LLMs) by focusing on diffusion-based LLMs (dLLMs), which enable parallel token generation via iterative denoising. However, existing parallel decoding strategies rely on fixed heuristics (e.g., confidence thresholds) that fail to adapt to input-specific characteristics, leading to suboptimal speed–quality trade-offs. The authors propose Learn2PD (“Learning to Parallel Decode”): a lightweight filter model trained in a post-training phase (requiring only minutes of GPU time) to predict whether a token’s current prediction matches the final output, thereby approximating an ideal “Extremely Greedy Parallel” (EGP) oracle that unmasks tokens only when predictions are correct—avoiding redundant re-decoding. Additionally, they introduce an End-of-Text Prediction (EoTP) mechanism to detect sequence termination and skip decoding of padding tokens. Evaluated on the LLaDA benchmark (GSM8K, MATH, HumanEval, MBPP), Learn2PD + EoTP achieves 22.58× speedup with no performance loss, and 57.51× with KV-Cache at minimal accuracy cost.

**Strengths:**

The method is elegantly simple and highly efficient: the filter model uses a two-layer MLP that takes block-level confidence scores as input and outputs a binary decision (remask or not) via a sigmoid-thresholded logit . Its computational overhead is negligible (<1% of inference time), avoiding complex architectures or task-specific engineering—demonstrating the principle that “simplicity works.” Moreover, the approach is highly reproducible: pseudocode is clearly provided, and inference requires no ground-truth labels, making it well-suited for open-source release and practical deployment.

**Weaknesses:**

- **Limited model and architecture coverage**: The method is evaluated exclusively on LLaDA (8B-Instruct) and not tested on other dLLMs such as DiffuLLaMA (Gong et al., 2025) or Dream (Ye et al., 2025), raising concerns about generalizability and model dependency.

- **Insufficient task and baseline comparisons**: The empirical analysis (e.g., Figure 2) focuses only on subsets like GSM8K and HumanEval, lacking comprehensive evaluation across the full LLaDA benchmark suite. Moreover, comparisons are limited to the official LLaDA baselines, omitting key alternatives such as confidence-threshold-based or slow-fast sampling strategies.

- **Oracle claims and training robustness**: The paper claims EGP achieves 15–20× optimal speedup, yet Figure 4 only reports a median of 2 steps per block without worst-case (lower-bound) analysis. Additionally, while the post-training phase is said to take “minutes,” there is no ablation on hyperparameter sensitivity (e.g., learning rate, threshold) or convergence behavior.

**Questions:**

- **Model details**: What are the exact specifications of \( f_\theta \)? Hidden dimension size? Activation function (e.g., ReLU)? Could incorporating confidence signals from more than one block improve performance? Additional experiments would help clarify design choices.

- **Transferability**: How would Learn2PD adapt to non-LLaDA dLLMs like Dream? Does retraining the filter incur prohibitive costs? What is the cross-task generalization gap (e.g., train on GSM8K, test on MATH)?

- **Robustness**: How does the method perform under low-resource settings (e.g., zero-shot) or with adversarial/noisy prompts? Furthermore, the “re-decoding gap” distribution in Figure 2 is based on only 10 samples—what are the mean and standard deviation across the full dataset?

---

> ### Author Response · Authors · 2025-11-24
> **Response to Reviewer pSk7**
>
> We thank reviewer pSk7 for the insightful and valuable comments. We sincerely hope that our rebuttal has addressed your concerns. If there are any remaining issues or points that are unclear, we would be grateful to receive further feedback and will gladly continue to respond and improve the work accordingly.
>
> **1. W1: Limited model and architecture coverage: The method is evaluated exclusively on LLaDA (8B-Instruct) and not tested on other dLLMs such as DiffuLLaMA (Gong et al., 2025) or Dream (Ye et al., 2025), raising concerns about generalizability and model dependency.**
>
> **Reply:**  Thanks for this suggestion. We have conducted GSM8K experiments on Dream under generation lengths of 256 and 1024. The results show that Learn2PD achieves 1.70× and 5.84× speedups over the vanilla Dream model at generation lengths 256 and 1024, respectively. Furthermore, EoTP improves the speedup to 10.99× at generation length 1024. These results suggest that both Learn2PD and EoTP are not limited to the LLaDA model and can transfer effectively to causal diffusion dLLMs such as Dream. We have updated the manuscript to include these results in Appendix and highlighted the corresponding entries in red.
>
> | Gen Length | Methods       | TPS  | Speed  | Accuracy   |
> |-----------|--------------|-------|--------|---------|
> | 256       | Dream           | 2.38  | 1.00×  | 0.746  |
> |           | + Learn2PD | 4.04 | 1.70×  | 0.745   |
> |           | Learn2PD & EoTP    | 4.73 | 1.99× | 0.734 |
> | 1024       | Dream          | 0.38 | 1.00×  | 0.719  |
> |           | + Learn2PD  | 2.19 | 5.84×  | 0.721   |
> |           | Learn2PD & EoTP    | 4.12 | **10.99×** | 0.731 |
>
> **2.W2: Insufficient task and baseline comparisons: The empirical analysis (e.g., Figure 2) focuses only on subsets like GSM8K and HumanEval, lacking comprehensive evaluation across the full LLaDA benchmark suite. Moreover, comparisons are limited to the official LLaDA baselines, omitting key alternatives such as confidence-threshold-based or slow-fast sampling strategies.**
>
> **Reply:** Thank you for this comment. Our empirical analysis is intended primarily as an illustrative example to highlight the limitations of existing dLLMs. Therefore, to save space, we only show results on GSM8K and HumanEval. A more comprehensive evaluation across the full LLaDA benchmark suite is provided in Section 4 (Experiments).
>
> In addition, we compare our method against other strategies. Their performance is displayed directly in their papers. We do not re-implement their methods.
>
> | Methods                    | GSM8K  | MATH   | HumanEval | MBPP     |
> |-------------------------|-----------|----------|---------------|-----------|
> | Fast-dLLM-Parallel [1]   | 2.5×       |  2.7×    | 3.3×           | 4.1×  |
> | SlowFast Sampling [2]   | 3.2×     | 2.2×      | 3.2×         | 2.7×       |
> | Learn2PD (ours)        | **4.1×**  | **3.2×** | **3.5×**        | **4.8×**  |
> All methods achieve accuracy scores that closely match those of the original LLaDA.
>
> [1] Wu, Chengyue, et al. "Fast-dLLM: Training-free Acceleration of Diffusion LLM by Enabling KV Cache and Parallel Decoding." arXiv (2025). https://arxiv.org/abs/2505.22618.
>
> [2] Wei, Qingyan, et al. "Accelerating Diffusion Large Language Models with SlowFast Sampling: The Three Golden Principles." arXiv (2025). https://arxiv.org/abs/2506.10848.
>
> **3.W3: Oracle claims and training robustness: The paper claims EGP achieves 15–20× optimal speedup yet Figure 4 only reports a median of 2 steps per block without worst-case (lower-bound) analysis. Additionally, while the post-training phase is said to take “minutes,” there is no ablation on hyperparameter sensitivity (e.g., learning rate, threshold) or convergence behavior.**
>
> **Reply:** Thank you for the insightful comments. We address them in two parts:
>
> 1.	If your question is about the details of EGP: Our EGP result is intended as an oracle-style, idealized analysis rather than a practically achievable speedup. In this oracle setting, the median number of inference steps per block is 2, which corresponds to an ideal speedup of approximately 32/2 = 16×, consistent with the “15–20×” range mentioned in the paper. The purpose of this experiment is not to present EGP as a deployable method, but to demonstrate that there is substantial space for accelerating inference process. For this reason, we did not provide a detailed quantitative study of its practical performance. If this is not what you were referring to, could you please clarify your question in more detail?
>
> 2.	Regarding the filter model, we include an ablation study on the threshold in Section 4.4. As we mentioned earlier, this task is quite simple, so the performance was quite insensitive to reasonable variations in the learning rate. The model can always eventually converge.

---

> ### Author Response · Authors · 2025-11-24
> **Response to Reviewer pSk7 (continue)**
>
> **4.Q1: Model details: What are the exact specifications of ( f_\theta )? Hidden dimension size? Activation function (e.g., ReLU)? Could incorporating confidence signals from more than one block improve performance? Additional experiments would help clarify design choices.**
>
> **Reply:** Thanks for this insightful comment. In our implementation, f_\theta is a simple two-layer MLP with ReLU activation, instantiated as a 32-dimensional input mapped to a 32-dimensional hidden layer and then back to 32 dimensions. In the existing model, a semi-autoregressive decoding scheme is adopted to improve accuracy, where decoding is performed at the block level. Our method is built on the same block size of 32, which has been a default choice in prior work. [1] We also compared 1-layer, 2-layer, and 4-layer MLPs and selected the optimal number of layers (2) for our filter model.
>
> [1] Nie, Shen, et al. "Large Language Diffusion Models." arXiv (2025). https://arxiv.org/abs/2502.09992.
>
> **5.Q2: Transferability: How would Learn2PD adapt to non-LLaDA dLLMs like Dream? Does retraining the filter incur prohibitive costs? What is the cross-task generalization gap (e.g., train on GSM8K, test on MATH)?**
>
> **Reply:** We appreciate this helpful comment. We provide the results on Dream in the response of Weakness 1. The training cost of filter model is only several minutes, so it would not incur prohibitive costs. To assess generalization across tasks, the current filter model is trained on FLAN dataset. Thus, a cross-task generalization gap is already reflected in the reported results. And if the filter is trained and tested on more closely matched datasets, its performance would further improve.
>
> **6.Q3: Robustness: How does the method perform under low-resource settings (e.g., zero-shot) or with adversarial/noisy prompts? Furthermore, the “re-decoding gap” distribution in Figure 2 is based on only 10 samples—what are the mean and standard deviation across the full dataset?**
>
> **Reply:** Thank you for this comment. In our current experiments, the filter does not require any task-specific fine-tuning for each benchmark (it is trained on 2640 FLAN samples and evaluated on GSM8K/MATH, etc.), which already reflects the zero-shot performance. For Figure 2, we analyze repetitive decoding steps only to demonstrate the potential for accelerating inference by reducing such redundancy. Therefore, we do not provide a full quantitative analysis of repetitive decoding steps over the entire dataset.

---

### Official Review · Reviewer_Y2jx · 2025-10-31

**Soundness:** 2
**Presentation:** 3
**Contribution:** 3
**Rating:** 6
**Confidence:** 2

**Summary:**

This paper proposes Learn2PD, a dynamic re-masking method that enables more efficient generation in diffusion language models. A learnable filter model is employed to achieve substantial acceleration in generation performance.

**Strengths:**

1. The paper presents a clear motivation, employing deep learning techniques to design a more principled unmasking mechanism.

2. The proposed Filter model is lightweight and easy to obtain.

**Weaknesses:**

1. The main experiments are conducted on the LLaDA model, and the effectiveness of the proposed method on causal diffusion models such as Dream remains to be further verified.

2. There is concern about the method’s performance on finer-grained blocks in mainstream block-level diffusion models.

**Questions:**

1. The proposed filter adopts a pure MLP architecture, which can be understood as a form of confidence-based feature engineering. Have you attempted to incorporate context-aware structures such as attention mechanisms?

2. There is a typo: on line 326, it should be 256 and 1024.

---

> ### Author Response · Authors · 2025-11-24
> **Response to Reviewer Y2jx**
>
> We thank reviewer Y2jx for the insightful and valuable comments. We sincerely hope that our rebuttal has addressed your concerns. If there are any remaining issues or points that are unclear, we would be grateful to receive further feedback and will gladly continue to respond and improve the work accordingly.
>
> **1. W1: The main experiments are conducted on the LLaDA model, and the effectiveness of the proposed method on causal diffusion models such as Dream remains to be further verified.**
>
> **Reply:**
> We appreciate this comment. We have conducted GSM8K experiments on Dream under generation lengths of 256 and 1024. The results show that Learn2PD achieves 1.70× and 5.84× speedups over the vanilla Dream model at generation lengths 256 and 1024, respectively. Furthermore, EoTP improves the speedup to 10.99× at generation length 1024. These results suggest that both Learn2PD and EoTP are not limited to the LLaDA model and can transfer effectively to causal diffusion dLLMs such as Dream. We have updated the manuscript to include these results in Appendix and highlighted the corresponding entries in red.
>
> | Gen Length | Methods       | TPS  | Speed  | Accuracy   |
> |-----------|--------------|-------|--------|---------|
> | 256       | Dream           | 2.38  | 1.00×  | 0.746  |
> |           | + Learn2PD | 4.04 | 1.70×  | 0.745   |
> |           | Learn2PD & EoTP    | 4.73 | 1.99× | 0.734 |
> | 1024       | Dream          | 0.38 | 1.00×  | 0.719  |
> |           | + Learn2PD  | 2.19 | 5.84×  | 0.721   |
> |           | Learn2PD & EoTP    | 4.12 | **10.99×** | 0.731 |
>
> **2. W2: There is concern about the method’s performance on finer-grained blocks in mainstream block-level diffusion models.**
>
> **Reply:** We appreciate this helpful comment. We have evaluated our method on finer-grained blocks with a block size of 16. The results on GSM8K show that, at generation lengths 256 and 1024, Learn2PD still achieves 3.96× and 9.83× speedups, respectively. These results demonstrate that our methods remain effective under finer-grained block-level settings. For the main experiments, we adopt a block size of 32, which is the default choice in existing block-level diffusion models. [1]
>
> | Block Size | Gen Length | Methods       | TPS  | Speed  | Accuracy    |
> |-----------|--------------|--------------|-------|--------|---------|
> | 16       | 256 |  LLaDA           | 3.27  | 1.00×  | 0.7885  |
> |           |         |  + Learn2PD | 12.97 | 3.96×  | 0.7892   |
> |           |         |   Learn2PD & EoTP | 14.93 | 4.56×  | 0.7839   |
> |           | 1024 |  LLaDA | 0.57 | 1.00×  | 0.7665   |
> |           |         |  + Learn2PD | 5.58 | 9.83×  | 0.7870   |
> |           |         |   Learn2PD & EoTP | 13.69 | 24.10×  | 0.7945   |
>
> [1] Nie, Shen, et al. "Large Language Diffusion Models." arXiv (2025). https://arxiv.org/abs/2502.09992.
>
> **3. Q1: The proposed filter adopts a pure MLP architecture, which can be understood as a form of confidence-based feature engineering. Have you attempted to incorporate context-aware structures such as attention mechanisms?**
>
> **Reply:** Thank you for your suggestion. The goal of our filter model is essentially a binary classification task that maps one 32-dimensional vector to another 32-dimensional vector. We firstly tested simple MLP architectures and found them capable of handling this task effectively. Therefore, we adopt a pure MLP architecture for the filter model. In particular, we compared 1-layer, 2-layer, and 4-layer MLPs and observed that the 2-layer model performed best, which further indicates that the task is relatively simple. Moreover, to maximize overall speedup, we want the filter model to introduce as little overhead as possible, which also favors a lightweight design. For these reasons, we adopt a simple 2-layer MLP as the filter model.
>
> **4. Q2: There is a typo: on line 326, it should be 256 and 1024.**
>
> **Reply:** Thank you for your suggestions. We have corrected this and highlighted it in the revised version.

---

> > ### Comment · Reviewer_Y2jx · 2025-11-27
> >
> > Thank you for the explanation. I will stand by my initial assessment.

---

### Meta-Review · Area_Chair_GhZa · 2026-01-08

**Summary:**

This paper proposes Learn2PD, a learnable and lightweight parallel decoding strategy for diffusion-based large language models, complemented by an End-of-Text Prediction mechanism. The core idea is to replace fixed, heuristic-based unmasking rules with a small neural filter that predicts whether a token’s current prediction already matches its final output, enabling adaptive and efficient parallel decoding. The problem is timely and important, as inference efficiency remains a key bottleneck for dLLMs.

Reviewers generally agree that the paper is well motivated and addresses a relevant and nontrivial problem in dLLM inference. The proposed method is consistently praised for its simplicity, elegance, and practicality: the filter model is extremely lightweight, easy to train, and introduces negligible inference overhead, yet yields substantial speedups. Experimental results demonstrate impressive acceleration particularly when combined with EoTP and KV-cache–based techniques.

Reviewers also raise moderate concerns about soundness, generality, and empirical coverage, which place the paper near the borderline. Most scores cluster around “fair” to “good” for soundness and contribution, with ratings ranging from marginally below to marginally above the acceptance threshold. Detailed concerns are:
1. Evaluation breadth: While the rebuttal adds Dream results, the overall evaluation still centers heavily on LLaDA, and comparisons to alternative decoding strategies are sometimes indirect (relying on reported numbers rather than re-implementations).
2. Clarity and presentation: Multiple reviewers found parts of the paper (e.g., Section 3.1.2 and Figures 2 and 4) hard to understand, though these issues were addressed in the revision.
3. Soundness and analysis depth: Some claims (e.g., oracle EGP speedups) are idealized and lack worst-case or robustness analysis. Hyperparameter sensitivity and robustness to distribution shifts are discussed but not extensively quantified.
4. Generalization and robustness: While transfer to Dream is now demonstrated, questions remain about cross-task generalization, behavior under noisy or adversarial inputs, and performance in lower-resource settings.

The authors provided thorough and constructive rebuttals, adding new experiments, clarifying misunderstandings, and improving figures and text. These responses address most of the reviewers’ concerns, particularly around generalization, inference overhead, and clarity. Some broader concerns about evaluation scope and robustness remain partially addressed but are not uncommon for a first introduction of a new decoding framework.

Overall, the submission presents a promising and practically impactful idea for accelerating diffusion LLMs, with a strong simplicity–performance trade-off. While the paper has some limitations in evaluation breadth and analysis depth, the rebuttal alleviates several key concerns. The consensus places the paper on the borderline, and I am slightly leaning towards acceptance if there is space.

**Reviewer Concerns:**

see above

**Reviewer Scores:**

see above

---

### Decision · Program_Chairs · 2026-01-26

Accept (Poster)